# Horse immunization with short-chain consensus α-neurotoxin generates antibodies against broad spectrum of elapid venomous species

Guillermo de la Rosa [1,2], Felipe Olvera[1], Irving G. Archundia[1], Bruno Lomonte [3], Alejandro Alagón[1] & Gerardo Corzo [1]

Antivenoms are fundamental in the therapy for snakebites. In elapid venoms, there are toxins, e.g. short-chain α-neurotoxins, which are quite abundant, highly toxic, and consequently play a major role in envenomation processes. The core problem is that such α-neurotoxins are weakly immunogenic, and many current elapid antivenoms show low reactivity towards them. We have previously developed a recombinant consensus short-chain α-neurotoxin (ScNtx) based on sequences from the most lethal elapid venoms from America, Africa, Asia, and Oceania. Here we report that an antivenom generated by immunizing horses with ScNtx can successfully neutralize the lethality of pure recombinant and native short-chain α-neurotoxins, as well as whole neurotoxic elapid venoms from diverse genera such as *Micrurus*, *Dendroaspis*, *Naja*, *Walterinnesia*, *Ophiophagus* and *Hydrophis*. These results provide a proof-of-principle for using recombinant proteins with rationally designed consensus sequences as universal immunogens for developing next-generation antivenom with higher effectiveness and broader neutralizing capacity.

[1] Departamento de Medicina Molecular y Bioprocesos, Instituto de Biotecnología, Universidad Nacional Autónoma de México – UNAM, Apartado Postal 510-3, Cuernavaca Morelos 61500, Mexico. [2] The Donnelly Centre for Cellular and Biomolecular Research, University of Toronto, Toronto, ON M5S3E1, Canada. [3] Instituto Clodomiro Picado, Universidad de Costa Rica, San José 11501, Costa Rica. Correspondence and requests for materials should be addressed to G.d.l R. (email: guillermo.delarosa.h@gmail.com) or to G.C. (email: corzo@ibt.unam.mx)

Snakebite is one the most neglected diseases, especially in the poorest tropical countries near the Equator. Literature analysis based on statistical estimates shows that up to 5.5 million snakebites could occur every year, yielding to more than 100,000 deaths worldwide[1]. The only venom-specific life-saving treatment proved as effective and recommended by the World Health Organization (WHO) is the timely parenteral application of snake antivenom[2]. Essentially, the active principle of anti-venoms is a polyclonal mixture of immunoglobulins, or fragments thereof, like Fab or F(ab′)$_2$. These are typically derived from the sera of hyper-immunized animals, mainly horses, and are able to neutralize the different venom toxins to prevent their deleterious effects[3].

In antivenom design and production, venoms from snakes responsible for causing high morbidity and mortality are commonly selected as immunogens[3]. In compliance with the host immune system, both toxic and non-toxic venom components elicit an antibody response; as a result, antivenoms contain collections of antibodies against both relevant and non-relevant components, which can affect antivenom efficacy. Postsynaptic α-neurotoxins are one of the main toxic elements in elapid venoms and the most poorly recognized components by current antivenoms[4–6], despite being rather abundant protein components in venoms used as immunogens.

α-Neurotoxins are classified as type I (short-chain), type II (long-chain), and non-conventional neurotoxins[7]. Short-chain α-neurotoxins (60–62 amino acids) have been associated with the high toxicity of many elapid venoms. They bind to the nicotinic acetylcholine receptors (nAChR) blocking neurotransmitter binding. Accordingly, they cause non-depolarizing blockade and consequently abolish neurotransmission, resembling curare-mimetic effects[8]. Under an elapid snakebite scenario, therefore, an effective anti-elapid therapy should have a collection of IgGs, F(ab′)$_2$, or Fab fragments able to properly neutralize α-neurotoxins in order to prevent or reverse postsynaptic neurotoxicity caused by these curare-mimetic toxins[9].

Aiming to develop complementary strategies to improve the antibody response and cross-recognition towards short-chain α-neurotoxins, our previous work focused on the design, recombinant expression, and biochemical characterization of a consensus type I α-neurotoxin with generic traits, here called ScNtx[10]. In this study, the ScNtx is used as an immunogen in horses, which are the preferred animal used for production of snake antivenoms

available on the market worldwide. The resulting anti-ScNtx experimental antivenom (EAv) efficacy and species coverage, expressed as median effective dose (ED$_{50}$), are systematically evaluated. Thus, our goal is to determine the extent of protection provided by this antivenom in mouse lethality tests against the challenge of isolated recombinant type I neurotoxins, as a proof of concept, and also against whole elapid venoms from snakes considered of highest medical importance in the Americas, Africa, Asia, and Oceania. Our results strongly suggest that a consensus α-neurotoxin as a rational-based immunogen in the production of antivenoms against neurotoxic elapid venoms could result in a product with a wide spectrum of specificity, efficacy, and affordability.

## Results

**ScNtx as immunogen.** In order to better understand the role of type I α-neurotoxins within the overall lethality of whole elapid venoms, we developed a horse-derived antivenom using a biologically active type I consensus α-neurotoxin, ScNtx, as a "universal" immunogen. The ScNtx was designed to show better antigenic properties (high therapeutic antibody titers) and thus to produce better antivenoms. We found that the group of three horses, immunized in a multi-site manner with increasing ScNtx doses (from 10 to 400 μg), produced antibodies that recognized the homologous immunogen in enzyme-linked immunosorbent assay (ELISA). Serum analysis shows that the response was rising over 210 days, meaning that increasing doses of ScNtx were efficient in inducing antibody production in these large animals. The titer of antibodies had a tendency to increase in response to repeated injections of the neurotoxin; nonetheless, ScNtx evoked a different response in the three animals (Fig. 1), one of them reaching a titer as high as 18,000.

**Lethal potencies.** The efficient assessment of antivenoms (ED$_{50}$) is based on their ability to neutralize the lethal effect of snake venoms (lethal dose, 50% (LD$_{50}$)). Therefore, we first determined the LD$_{50}$ for all toxins and venoms studied in this work (Table 1). In total, we used 4 short-chain α-neurotoxins and 29 elapid snake venoms from species distributed in different world regions. The short-chain α-neurotoxins had intravenous (IV) LD$_{50}$s ranging from 1.2 to 19.0 μg/mouse. Concerning the elapid venoms, those from the *Micrurus* genus had LD$_{50}$s ranging from 3.8 μg/mouse (*Micrurus browni*) to 15.0 μg/mouse (*Micrurus tener tener*); venoms from *Dendroaspis* ranged from 5.3 to 17.0 μg/mouse; the LD$_{50}$s of the venoms from *Naja* genus were from 0.9 (*Naja haje* from Morocco) to 22.7 μg/mouse (*Naja mossambica*); venoms from *Pseudechis* were from 6.9 to 8.9 μg/mouse, and finally, the LD$_{50}$s of venoms from *Ophiophagus hannah*, *Walterinnesia aegyptia*, *Oxyuranus scutellatus*, and sea snake *Hydrophis (Pelamis) platura* were 11.5, 4.9, 0.7, and 3.9 μg/mouse, respectively.

**ScNtx elicits neutralizing antisera.** First, we assessed the individual response and maturation of the immune response based on the neutralization potency of three serum samples collected throughout the immunization period of the horses, corresponding to days 98, 147, and 206. Results presented in Fig. 2 revealed that animals exhibited differences in neutralization potency against α-neurotoxins. In general, effective doses (ED$_{50}$) were higher in samples from day 98 and lower from day 206, indicating an increase in neutralizing potency over time.

Second, we developed two lots of EAvs called Lot #1 and Lot #2. Lot #1 contains antibodies from sera pooled from horse 2 at days 147 and 205, and Lot #2 has antibodies from the day 708. Both lots have a protein content of 50 mg/mL. Lot #1 was

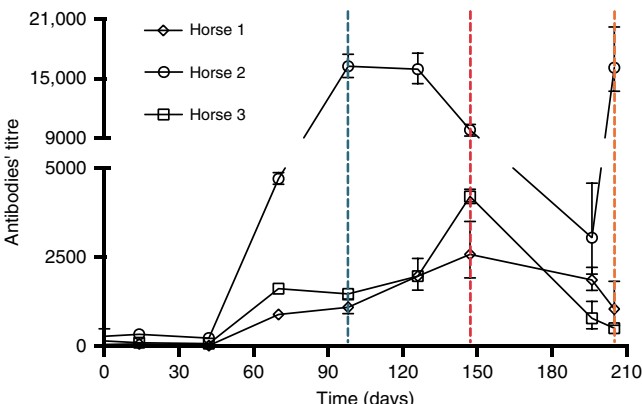

**Fig. 1** Time course of the specific antibody response of horses against the consensus short-chain α-neurotoxin (ScNtx). Horse hyperimmune sera raised against the ScNtx were titrated by enzyme-linked immunosorbent assay (ELISA). Dashed lines indicate samples used for neutralization tests, corresponding to days 98 (blue), 147 (red), and 205 (orange) of the immunization. Points represent mean ± SD of triplicate wells of the ELISA

**Table 1 Median $LD_{50}$ and $ED_{50}$ values of α-neurotoxins or elapid venoms, as well as median values of antivenom (EAv)**

| α-Neurotoxin/ venom | $LD_{50}$ in µg/mice (95% CI) | Horse EAv $ED_{50}$ (µL/mice) (95% CI), Lot #1 | Horse EAv $ED_{50}$ (µL/mice) (95% CI), Lot #2 | Average mg toxin or venom/mL EAv | Average $LD_{50}$ of toxin or venom/ mL EAv |
|---|---|---|---|---|---|
| α-Neurotoxins | | | | | |
| ScNtx | 3.9 (3.8–3.9) | 49.1 (47.8–50.4) | 40–45 | 0.23 | 61.1 |
| MlatA1 | 19 (18–20) | 45.4 (44.1–46.8) | ND | 1.25 | 66.1 |
| r.D.H | 3 (2.8–3.2) | 31.7 (31.7–31.8) | ND | 0.28 | 94.6 |
| P01424[a] | 1.2 (1.1–1.3) | 34.3 (33.7–34.9) | ND | 0.17 | 145.7 |
| Venoms | | | | | |
| Micrurus browni | 3.8 (3.5–4.1) | NN | ND | – | – |
| Micrurus diastema | 6 (5.5–6.5) | 191.5 (188.7–194.3) | NN | 0.09 | 15.6 |
| Micrurus distans | 12 (10–14) | NN | ND | – | – |
| Micrurus fulvius | 4 (3.8–4.2) | NN | ND | – | – |
| Micrurus laticorallis | 10 (9.1–11.4) | 171.1 (166.0–175.0) | NN | 0.17 | 17.5 |
| Micrurus nigrocinctus | 7 (5–9) | 56.5 (54.8–58.1) | NN | 0.37 | 53.1 |
| Micrurus surinamensis | 10 (9.5–10.5) | 51.2 (49.7–52.7) | 25.6 (20.0–30.0) | 0.78 | 78.1 |
| Micrurus tener tener | 15 (13–17) | NN | ND | – | – |
| Dendroaspis angusticeps | 17 (16–18) | 178.0 (171.3–185.4) | NN | 0.28 | 16.8 |
| Dendroaspis polylepis | 5.3 (5–5.5) | 149.1 (141.2–157.3) | NN | 0.16 | 20.1 |
| Dendroaspis viridis | 12 (11.9–12.2) | NN | ND | – | – |
| Naja atra | 8 (7.7–8.2) | 37.5 (35.5–39.5) | 34.2 (28.5–41.1) | 0.67 | 83.0 |
| Naja haje[a] | 0.9 (0.8–1) | 77.4 (76.6–78.2) | 72.9 (63.9–81.0) | 0.04 | 40.0 |
| Naja kaouthia | 3.9 (3.7–4.2) | 50.0 (47.0–53.0) | >400 | 0.23 | 60.0 |
| Naja katiensis | 20.5 (18.4–22.7) | ND | >400 | – | – |
| Naja oxiana | 8 (7.0–9.8) | ND | 24.1 (20.1–28.5) | 0.99[b] | 124.4[b] |
| Naja melanoleuca | 6.5 (6.4–6.6) | 72.3 (69.9–74.7) | 306.0 (301.0–311.0) | 0.11 | 15.8 |
| Naja mossambica | 22.7 (19.5–25.8) | NN | NN | – | – |
| Naja naja naja | 9.4 (9.3–9.6) | 65.5 (63.1–67.7) | ND | 0.43 | 45.8 |
| Naja nigricollis | 18 (17–19) | NN | ND | – | – |
| Naja nivea | 8.2 (8–8.4) | 25 (20–30) | 106.0 (92.5–121.5) | 0.38 | 46.7 |
| Naja nubiae | 8.3 (8.1–8.4) | 173.7 (169.1–178.4) | ND | 0.14 | 17.3 |
| Naja pallida | 17 (17.8–17.2) | NN | ND | – | – |
| Ophiophagus hannah | 11.5 (10.9–12.2) | 47.9 (44.1–52.1) | 74.9 (73.2–76.7) | 0.56 | 48.8 |
| Hydrophis platura | 3.9 (0.8–6.4)[c] | 40–66 | ND | No <0.17 | No <45.4 |
| Walterinnesia aegyptia | 4.9 (3.9–5.8) | 55.7 (53.1–58.5) | 44.5 (40.0–49.5) | 0.29 | 59.8 |
| Oxyuranus scutellatus | 0.7 (0.6–0.8) | ND | NN | – | – |
| Pseudechis australis | 6.9 (5.6–8.6) | ND | NN | – | – |
| Pseudechis colleti | 8.9 (8.4–9.5) | ND | NN | – | – |

Lot #1 and Lot #2 have a protein content of 50 mg/mL; for Lot #2 >400 denotes that neutralization was found but no 100% of survival
95% CI 95% confidence intervals (shown within parentheses), $ED_{50}$ median effective dose, $LD_{50}$ lethal dose, EAv experimental antivirus, NN no neutralization at a maximum level of 200 µL/mouse for Lot #1, and 400 µL for Lot #2 (100% lethality), ND not determined
[a]5 × $LD_{50}$ were used for $ED_{50}$ determination in this case. All other values were obtained with a venom challenge of 3 × $LD_{50}$
[b]Based on Lot #2
[c]Value obtained from ref. [45]

comprehensively evaluated, while Lot #2 was used only used for some venoms.

**ScNtx antivenom**. The vast array of pathophysiological symptoms exhibited by elapid envenomation are related to presynaptic- and postsynaptic-acting toxins. Venomics and conventional research have unveiled many predominant presynaptic and predominant postsynaptic elapid venoms. Thus, we aimed to assess the extent of protection of our EAv against short-chain α-neurotoxins and well-known elapid venoms from the Americas, Africa, Asia, and Oceania.

**Efficacy of ScNtx antivenom against purified α-neurotoxins**. The EAv Lot #1 neutralized the lethal effect of purified type I α-neurotoxins. The $ED_{50}$ values determined for this preparation, against α-neurotoxins, are expressed in µL/mouse and listed in Table 1. This anti-ScNtx preparation neutralized the lethality of all the purified α-neurotoxins, with $ED_{50}$ values around 40 µL/ mouse. EAv had an $ED_{50} = 49.1$ µL/mouse against the ScNtx, meaning 230 µg of ScNtx neutralized by 1 mL/EAv or 230 µg of toxin/50 mg of immunoglobulins. Lot #2 had a similar range of potency. Similarly, as shown in Table 1, EAv Lot #1 had an $ED_{50}$ $= 45.1$ µL/mouse (1.2 mg/mL EAv) and $ED_{50} = 31.7$ µL/mouse (280 µg/mL EAv) against recombinant MlatA1 and r.D.H,

 3

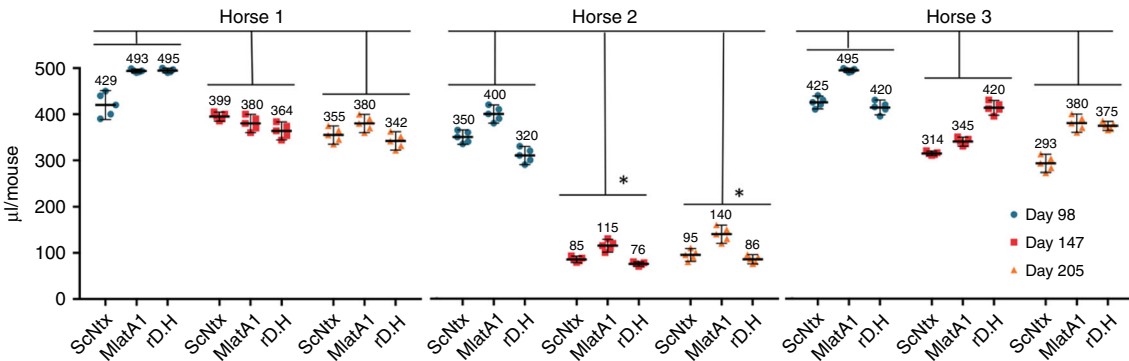

**Fig. 2** Neutralization potencies, plotted as median effective dose ($ED_{50}$). Serum samples were obtained from individual horses immunized with the consensus short-chain α-neurotoxin (ScNtx) at days 98, 147, and 205. $ED_{50}$ values are expressed in μL/mouse (volume of anti-ScNtx able to neutralize 3 × $LD_{50}$ (lethal dose, 50%) of purified short-chain neurotoxins ScNtx, MlatA1 or rD.H.). Values were estimated in groups of five mice, injected by the intravenous route. Error bars represent the 95% confidence limits for the $ED_{50}$ value. The data obtained from the groups of "Horse" and "Day" of immunization were analyzed statistically by two-way analysis of variance (ANOVA) followed by paired Student's $t$ test. A $p$ value of <0.05 was considered significant, as indicated by *. Horse 2 produces antibodies with the best neutralization potencies from after the 147th day of immunization

| Toxin | Amino acid sequence | Identity (%) | Accession # |
|---|---|---|---|
| ScNtx | MICYNQQSSQPPTTKTCS--ETSCYKKTWRDHRGTIIERGCGCPKVKPGIKLHCCRTDKCNN | 100 | |
| rD.H | MICHNQQSSQPPTTKTCS--EGQCYKKTWRDHRGTIIERGCGCPTVKPGIHISCCASDKCNA | 85 | P86420 |
| MlatA1 | RICYNQQSSQPPTTKTCS--EGQCYKKTWRDHRGTIIERGCACPNVKPGIQISCCTSDKCNG | 84 | K9MCH1 |
| P01426 | LECHNQQSSQPPTTKTCP-GETNCYKKVWRDHRGTIIERGCGCPTVKPGIKLNCCTTDKCN | 83 | P01426 |
| P80548 | MICHNQQSSQPPTIKTCS--EGQCYKKTWRDHRGTISERGCGCPTVKPGIHISCCASDKCNA | 81 | P80548 |
| Atratoxin | LECHNQQSSQTPTTTKTCS-GETNCYKKWWSDHRGTIIERGCGCPKVKPGVNLNCCTTDRCNN | 79 | AAR33036 |
| MS1 | MICYNQQSTEPPTTKTCS--EGQCYKKTWRDHRGTIIERGCACPNVKPGVKISCCSSDKC | 79 | P86095 |
| P01424 | MECHNQQSSQPPTTKTCP-GETNCYKKQWSDHRGTIIERGCGCPSVKKGVKINCCTTDRCNN | 77 | P01424 |
| Pelamitoxin | MTCCNQQSSQPKTTTNCA--ESSCYKKTWSDHRGTRIERGCGCPQVKSGIKLECCHTNECNN | 75 | P62388 |
| W-III | FVCHNQQSSQPPTTTNCSGGENKCYKKQWSDHRGSITERGCGCPTVKKGIKLHCCTTEKCNN | 73 | C1IC47 |

Loop I — Loop II — Loop III

**Fig. 3** Multiple sequence alignment of the most lethal short-chain α-neurotoxins. ScNtx: Synthetic consensus α-neurotoxin[10]; rD.H: *Micrurus diastema* (recombinant)[53]; MlatA1: *Micrurus laticollaris* (recombinant)[52]; P01426: *Naja* spp.; P80548: *Micrurus nigrocinctus*[4]; Atratoxin: *Naja atra*; MS1: *Micrurus surinamensis*; P01424: *Naja melanoleuca*; pelamitoxin: *Hydrophis platura*; W-III: *Walterinnesia aegyptia*. Differences from the ScNtx are indicated by bold letters; cysteines and disulfide bridge arrangements are colored in blue

respectively, and an $ED_{50} = 34.3$ μL/mouse (170 μg/mL EAv) against the native type I α-neurotoxin, P01424.

**Efficacy of ScNtx antivenom against coral snake venoms.** *Micrurus fulvius, M. tener, M. browni,* and *M. distans* whole venoms are not neutralized by the EAv (efficient against type I α-neuro-toxins). On the other hand, EAv Lot #1 neutralized *M. laticollaris* ($ED_{50} = 171.0$ μL/mouse) and *M. diastema* ($ED_{50} = 191.0$ μL/mouse) venoms with higher doses. That means that 170 and 90 μg of each venom is neutralized by 1 mL of EAv (50 mg). Nonetheless, venoms from these species but obtained from different individuals were not neutralized by Lot #2. Similar findings were determined for the Central American coral snake *Micrurus nigrocinctus*. On the one hand, EAv Lot #1 was able to successfully neutralize its lethality ($ED_{50} = 56.6$ μL/mouse). On the other hand, EAv Lot #2 did not neutralize the lethality of a different individual. Finally, for *M. surinamensis* both lots of EAv were highly efficient in neutralizing the lethality of the venom from the same individual snake ($ED_{50} = 51.6$ and $25.6$ μL/mouse, respectively) (Table 1).

**Efficacy of ScNtx antivenom against cobra venoms.** To deter-mine the efficacy of the EAv on more complex elapid venoms, we performed neutralization assays on 13 different species of spitting and non-spitting cobras. In our findings, the lethality of venoms of the spitting cobras *Naja nigricollis, N. pallida, N. katiensis,* and

*N. mossambica* was not abolished by the EAv, unlike the *N. nubiae* venom ($ED_{50} = 173$ μL/mouse) (Table 1). We next addressed whether the EAv was effective against the non-spitting cobras. As a first finding, our data indicated that non-spitting cobras (*Naja*) show, in most cases, the most lethal venoms. We also found that they are the best neutralized by the EAv here characterized. For example, the *N. haje* venom used in this work had a $LD_{50} = 0.9$ μg/mouse, and its lethality was inhibited by the EAv Lot #1 ($ED_{50} = 77.4$ μL/mouse). Similar efficacy was found for the venoms from *N. melanoleuca* ($ED_{50} = 72.4$ μL/mouse) and *N. naja* ($ED_{50} = 65.5$ μL/mouse).

## Discussion

Snake venoms are an exquisite panoply of molecules, mostly proteins, contributing differentially in the pathophysiological effects in snakebites. Nonetheless, such complexity is expressed as a relatively small but important set of clinical manifestations, such as flaccid paralysis, local tissue damage, systemic myolysis, hemorrhage, and coagulation alterations, among others. For efficient immunotherapy against snakebite envenoming, proper neutralization of toxins playing important roles is required[11]. A key set of potential targets are the elapid α-neurotoxins, which cause flaccid paralysis leading to death by respiratory failure. However, one of the main drawbacks related to α-neurotoxins is their poor immunogenicity. Additionally, many works have

suggested the role of type I α-neurotoxins in the lethal potency of venoms stemming from α-neurotoxins' high toxicity, molecular action, and abundance[5,12].

Overall, the efficacy of the experimental antivenom (ScNtx antivenom) correlates with the presence of type I α-neurotoxins and some other members of the 3FTx family within the venoms. First, the neutralization of pure α-neurotoxins by the EAv suggest shared immunogenic determinants among these toxins. In fact, comparing their amino acid sequences, shown in Fig. 3, we qualitatively deduced a high conservation in two regions: from the first to the third cysteine (first loop) and from the second to the fourth cysteine (Cys) residue, which bears the region corresponding to the second loop. The latter, particularly, contains the amino acid residues involved in the toxin–receptor interaction. As expected, the conservation of some residues at the second loop among the ScNtx sequence and other 3FTxs such as fasciculins[13] and long-chain neurotoxins[13–15] agrees with the conclusion of some researchers that postulates this region as a conserved neutralization epitope on α-neurotoxins[16]. Henceforth, in-depth studies on mapping the interactions of ScNtx with its elicited horse antibodies would be revealing to this hypothesis.

*Micrurus* produces potent neurotoxic venoms that act on the neuromuscular junction. It has been demonstrated that there are two main protein families among them: phospholipases $A_2$ (PLA2s) and three-finger toxins (3FTxs). In a global view, the proportion of PLA2s vs. 3FTxs varies among North American and South American species[17]. For instance, venoms from *M. fulvius* (Florida) and *M. tener* (Texas) are PLA2 predominant, while *M. surinamensis* (South America) contains a 3FTx-rich venom almost devoid of PLA2 activity. For the former, it has been demonstrated that such PLA2s are the main lethal components in the mouse model of envenomation[18]. In the case of *M. nigrocinctus* (Central America), which contains a venom that is PLA2 predominant over 3FTxs[4], EAv was able to successfully neutralize its lethality ($ED_{50}$ = 56.6 μL/mouse) and had similar efficacy to some commercial antivenoms ranging from 63 to 123 μL/mouse (Coralmyn® and anti-Micrurus, respectively)[19]. We conclude that the α-neurotoxin fraction is highly relevant for its lethality. In this case such effect could be led by the main short-chain α-neurotoxin P80548, which presents 81% identity compared to the ScNtx, as shown in Fig. 3, which is also the most lethal toxin in this venom[4]. Similarly, both the EAv Lot #1 and Lot #2 were highly efficient in neutralizing the lethality of *M. surinamensis* (3FTx-rich venom) venom, which contains a large arsenal of α-neurotoxins that shares high identity in sequence to ScNtx (>74%, Fig. 3)[20].

The EAv is efficient against 3FTx-rich cobra (*Naja*) venoms. Cobra venoms contain 3FTxs, PLA2s, metalloproteinases (SVMPs), Cys-rich secretory proteins, and L-amino oxidases (LAAOs)[21,22] as principal protein families. Notwithstanding this set of toxins, most human envenomations result in neurological affectations and in some cases serious local injuries, which correlate with the predominance of 3FTxs, SVMPs, or PLA2s. For instance, *N. nigricollis*, *N. katiensis*, *N. pallida*, and *N. mossambica* are rich in cytotoxins/cardiotoxins, primarily, followed by PLA2s and metalloproteinases. These toxins cause local tissue necrosis and hemorrhage; therefore, the inefficacy of our EAv on these venoms agrees with their proteomics and distinctive clinical manifestation[22,23]. On the other hand, the EAv does neutralize 3FTx-predominant cobra venoms such as *N. nubiae* (type I neurotoxin content ~12.6%)[22], *N. haje* (3FTx = ~60%)[24,25], *N. melanoleuca* (3FTx = ~57%)[25,26]—one possible target could be the α-neurotoxin *P01424* ($LD_{50}$ = 0.9 μg/mouse) that greatly influences its lethality and that we showed is neutralized by the EAv—*N. naja* (3FTx = ~63.8%)[27]—which is one of the main species responsible for snakebite mortality in South-Asia—and

*Naja oxiana* (type I neurotoxin content >12%)[28]. The present study has only investigated the neutralization of the venom/toxin-induced lethality. Consequently, this work clearly has some limitations. The most important lies in the fact that we cannot demonstrate whether EAv cross-reacts with other 3FTxs, such as type II, and cytotoxin/cardiotoxins. However, our results indicate that cross-reaction might occur. For instance, antibodies to ScNtx in the EAv were able to protect from lethal doses of the venoms from *N. atra*, *N. nivea*, and *N. kaouthia* (3FTx = 56–78%), which are not only rich in cytotoxins (e.g., CTX)[29–31] but also in type II α-neurotoxins (e.g., α-cobratoxin P0139)[21–32]. It is well known that *N. atra* venom produces cytotoxicity resulting from the action of the group of toxins historically known as CTX[29,31], and likewise produces neurotoxicity caused mainly by type I α-neurotoxins (e.g., atratoxins) and muscarinic toxins (MTs), which are also considered to play important roles in the envenoming process. *Naja nivea* venom, in turn, displays important myotoxic, cardiotoxic, and neurotoxic activities[31]. These results reveal as a whole the specific role of type I α-neurotoxins vs. other toxic elements on the lethality of these venoms and an arguable cross-reactivity among other 3FTxs.

To test the role of a likely cross-reactivity among 3FTxs, we used the venom of Mambas. They produce venoms that contain a rich cocktail of rapid-acting neurotoxins[32]. Human envenomation by mambas can lead to neurotoxicity with no reports of local tissue damage[8]. Toxins acting on potassium channels (Kv), L-type calcium channels, acid-sensing ion channels[33], cholinergic receptors[34], and others acting on platelet aggregation[35] have been reported. Nonetheless, the evidence suggests that 3FTxs, such as MTs, fasciculins, and α-neurotoxins, coupled with presynaptic-acting Kunitz-type proteinase inhibitor-like (dendrotoxins) could be highly relevant for mamba venoms lethality[36,37]. Under an anti-α-neurotoxin view, neutralization of these ScNtxs was achieved for *Dendroaspis angusticeps* and *D. polylepis*, but *D. viridis* lethal effect remained unaltered. These results show how important the set of diverse neurotoxins (e.g., MTs, fasciculins, dendrotoxins, long-chain α-neurotoxins) are to the lethality of *Dendroaspis* and some *Naja* venoms. They also suggest that the neutralization of the α-neurotoxins, and the possible cross-reactivity, is not sufficient to provide protection since mice, even when they survived, in all cases showed envenomation symptoms.

Here, it is worth mentioning that long- and short-chain α-neurotoxins are equally important during elapid envenomation, and more studies are needed to decipher the pharmacokinetics and biochemistry of each one of these neurotoxins. Also, the use of mice as an animal model for interpreting neutralization of venoms that are rich in short- or long-chain α-neurotoxins has to be revised. Silva et al.[38] found differential susceptibility of human and rat nAChR to short-chain α-neurotoxins, but not to long-chain α-neurotoxins. This finding may represent an important challenge for the current assessment of the preclinical efficacy of antivenoms in mouse models.

Hence, further studies should take into consideration the possible combined action among mamba toxins in order to postulate additional antigens for immunization. Moreover, our results promote further research in order to have a deep insight about cross-reactivity and the variation in terms of efficacy between the different lots of EAvs against *Dendroaspis* venoms. In this regard, it is important to underline that such variation was not observed for all venoms. For instance, for the same batch of king cobra venom (*Ophiophagus hannah*) and black snake (*Walterinnesia aegyptia*) venom the $ED_{50}$s among EAv lots were similar. We know that these venoms are 3FTx predominant; nonetheless, their complexity is markedly different. King cobra venom, for example, contains LAAOs, SVMPs, and PLA2s[39]. Among the rich repertoire of 3FTxs, it has been proven that long-

chain α-neurotoxins play a key role in its lethality given the fact that specific antibodies produced against them attenuate the venom toxicity[40,41]. Importantly, other toxins, like the short-chain α-neurotoxins, even though they are less abundant, are substantially crucial for the venom lethality[42]. Our results support this observation, as we demonstrated that O. hannah lethality can be efficiently abolished by both lots of EAv, possibly by acting on both long-chain and short-chain neurotoxins. On the other hand, W. aegyptia venom, while less complex, is very active post-synaptically, causing life-threatening neurotoxic envenomation[43]. The most abundant toxins in this venom are short-chain α-neurotoxins, PLA2s, and Kunitz-type proteinase inhibitor-like proteins[43,44]. Nevertheless, only short-chain α-neurotoxins have been considered as targets in the development of a new anti-walterinnesia antivenom. For instance, the short-chain α-neuro-toxin $T_{III}$, which is the most abundant toxin in the venom, was used to develop an antivenom that neutralizes 50 $LD_{50}/mL$[44]; comparing this, 1 mL of EAv neutralizes 53 $LD_{50}s$, supporting the concept that W. aegyptia lethality is driven by its principal components, namely short-chained α-neurotoxins. Similarly, the most abundant and lethal component of the sea snake H. platura (3FTx = ~50%) venom is the α-neurotoxin known as pelami-toxin (P62388) and its proteoforms[45]. Owing to their high identity to the ScNtx, both $T_{III}$ and pelamitoxin are likely the targets of the neutralizing anti-ScNtx antibodies.

To question whether the EAv could neutralize or attenuate the lethality of elapid venoms from Oceania, we used the venom of two genera distributed in Australia and parts of New Guinea: Pseudechis australis, P. colleti, and Oxyuranus scutellatus. These venoms contain short-chain neurotoxins and they are also active at the presynaptic level like some Micrurus venoms[46]. None-theless, when tested, EAv did not have any effect on their leth-ality. Although these venoms cause neurotoxicity that leads to respiratory arrest, our result suggest that this might be caused, mainly, by other group of neurotoxins such as long-chain neu-rotoxins or presynaptically acting PLA2s rather than by short-chain neurotoxins. Alternatively, these venoms could contain short-chain neurotoxins that are more antigenically divergent from the consensus sequence immunogen here studied.

In conclusion, in this paper we have described the range of efficacy for a polyspecific horse-derived experimental antivenom capable of neutralizing medically important elapid venoms dis-tributed throughout the Americas, Africa, Middle East, and some regions of Asia. The evidence from this study has revealed the importance of short-chain α-neurotoxins in the lethality of sev-eral Micrurus, Naja, Dendroaspis, Walterinnesia, Ophiophagus, Oxyuranus, Pseudechis, and Hydrophis venoms.

Furthermore, due to the fact that snake venoms can present intra-species variability in their composition, recombinant proteins could be a constant source of active toxins (immunogens) for better antidotes. For this, it is clear that further work needs to be done to elucidate more medically relevant toxins. Correspondingly, it is important for future research to consider variation in venoms composition to understand how this variability could also influence both envenomation signs and antivenom efficacy[47,48]. The rational concept of using a recombinant neurotoxin with "universal" con-sensus sequence to produce a broad-specificity anti-elapid anti-venom provides the framework for a strategy to develop either animal-derived or recombinant antivenoms. We hope that our research will be useful in solving the difficulty of conceiving alter-native immunogens to create better therapeutic antivenoms with higher neutralizing potencies and broader coverages.

## Methods
**Venoms and toxins**. Venoms used in this study were from different sources. Certified, N. melanoleuca (715.030 and 307.150), N. katiensis (code not provided),

N. oxiana (911.040), N. pallida (316.000), N. nivea (524.010), N. nubiae (101.030), N. mossambica (505.010), N. nigricollis (105.030), N. haje (822.090), N. kaouthia (506.000), N. atra (920.100), D. polylepis (218.020), D. angusticeps (305.00), D. viridis (815.050), W. aegyptia (729.030), and O. hannah (923.090) venoms were from Latoxan (Valence, France); N. nivea (NNC019) was from SA Venom Sup-pliers (Louis Trichardt, South Africa); N. naja naja (4NN9001) from Ventoxin; the australian venoms Oxyuranus scutellatus (OSS0319EAS, Cooktown, Australia), P. australis (PA0319EAS, Coen Cape York, Australia), P. colleti (PC0319EAS, Vergemont, Australia) were acquired through "La Nauyaca" (UMA INE/CITES/ DFYFS-HERP-E-0003-MOR/98, Morelos, Mexico); venoms from M. fulvius (08.31.10) and M. tener tener (02.16.09) were from National Natural Toxins Research Center; and venoms from M. laticollaris (two individuals), M. browni, M. diastema (two individuals), M. distans, M. nigrocinctus (two individuals), M. surinamensis, and sea snake H. (Pelamis) platura[24] were obtained manually by milking multiple adult snakes at the "Instituto de Biotecnología" (UNAM, Mexico, permit SGPA/DGVS/010526/18) and "Instituto Clodomiro Picado" (UCR, Costa Rica, permit CICUA-021-17) serpentariums.

**Expression and purification of α-neurotoxins**. Stored pQE30/ScNtx[10], pQE30/ MlatA1[49], and pQE30/r.D.H[50] plasmids were transformed into Escherichia coli K-12-derived Origami cells for expression following the same conditions as in our previous work (for a more detailed description see de la Rosa et al.[10]). In short, expressed recombinant ScNtx, MlatA1 or r.D.H were submitted to a two-step purification process: by Ni-NTA (Ni-nitrilotriacetic acid) affinity chromatography according to the method of polyhistidine-tagged proteins user manual (Qiagen); and by reversed-phase high-performance liquid chromatography (RP-HPLC) (Agilent 1100 series; Agilent, CA) loading them onto an analytical $C_{18}$ column (4.6 × 250 mm², VYDAC®). Elution was carried out at 1 mL/min for 60 min by applying a gradient starting at 10% of an aqueous acetonitrile solution/0.1% tri-fluoroacetic acid (TFA) and ending at 60%. Similarly, in order to isolate the α-neurotoxin P014242 (UniProt access code) from N. melanoleuca, 2 mg of crude venom were fractionated by RP-HPLC using a gradient toward acetonitrile solu-tion/0.1% TFA maintaining 0% 5 min, 0–15% over 10 min, 15–45% over 60 min, 45–70% over 10 min, and 70–80 over 5 min[31]. Finally, all identities were confirmed by mass spectrometry.

**Animals**. Mice strain CD-1 (18–20 g) were purchased from Harlan Mexico and kept at animal facilities at the "Instituto de Biotecnología", Mexico, and also, they were provided and kept at animal facility at the "Instituto Clodomiro Picado", Costa Rica. Adult male horses (400–500 kg) were from Ranch "Ojo de Agua" (Puebla, Mexico). All animals received regular veterinary supervision and were maintained under good conditions and controlled environments. They received water and food ad libitum. Proper animal handling, in order to minimize distress and discomfort, was always conducted towards maximizing the animal welfare during experimentation according to Mexican legislation for the use of laboratory animals (Norma Oficial Mexicana, 1999, NOM-062-ZOO-1999). Additionally, internal animal handling was according to the Animal Care and Bioethics Com-mittee at the "Instituto de Biotecnología" (ethical approval CB/IBt/Project # 254) and "Instituto Clodomiro Picado" (ethical approval code: CICUA-021-17), which supervised and approved all animal experiments.

**Immunization, antisera collection, and sera fractionation**. Three horses were immunized by multi-site intradermal and subcutaneous route[51], starting with 5 μg and ending with 1000 μg of ScNtx per horse. The first immunization of 5 μg was performed in 0.5 mL phosphate-buffered saline plus 0.7 mL incomplete Freund's adjuvant (IFA) by intradermal route. Horses were boosted 14, 28, 42, 56, 70, 84, 98, 112, 126, 140, and 196 days later with 5, 5, 10, 20, 40, 80, 150, 300, 300, 500, 500, and 1000 μg of the ScNtx, alternating IFA and Alum. No adjuvant was used on the seventh and the last two immunizations. For all cases, horses were bled at weekly intervals until day 147 and the last two blood samples were taken at days 196 and 205. Then, after a 488-day hiatus, only horse 2 was boosted four times at weekly intervals with 300 (day 684), 500 (day 692), 1000 (day 700), and 1800 μg (day 708). IFA and Alum were used in these first two boosts and no adjuvants were used in the last two; a blood sample was taken 1 week after the last immunization. Animals' blood was allowed to clot at 37 °C for at least 2 h, chilled on ice for 1 h, and centrifuged at 4000 × g. Serum was collected and stored frozen at −20 °C until used.

Finally, horse immunoglobulins were obtained by caprylic (octanoic) acid method[52] described and recommended in the WHO guidelines for the production, control, and regulation of antivenoms[53]. Two different lots were prepared: Lot #1 contains immunoglobulins from days 147 and 205, and Lot #2 from day 708. Each Lot has a protein content of 50 mg/mL.

**Median $LD_{50}$ and $ED_{50}$**. Lethal potencies for toxins and whole venoms (in μg dry weight per mouse) were determined by calculating the $LD_{50}$, which is defined as the amount of venom that produces the death of 50% of the mice challenged. Briefly, five mice per group were injected by IV route and the $LD_{50}$ was obtained from the plot analysis of mice mortality (at 24 h after injection) vs. toxin or venom dose used. The $LD_{50}$ was expressed in μg/mouse. For neutralization experiments, 3 ×

$LD_{50}$ of either toxin or whole venom were pre-incubated 30 min at 37 °C with varying volumes of antiserum and then injected by IV route. After 24 h, median effective dose ($ED_{50}$) was calculated from the plot of survival percent (at 48 h) vs. antiserum dose, and it is defined as the volume of antiserum able to protect 50% of the mice challenged (five mice per group)[19]. Only for the native type I α-neurotoxin P014242, *N. haje* and *Oxyuranus scutellatus*, due to their high toxicity, $5 \times LD_{50}$ of each were used for $ED_{50}$ calculation. Control groups tested without antiserum or with pre-immune horse sera always resulted in 100% mortality; likewise, 400 μL (50 mg/mL) of purified horse pre-immune immunoglobulins showed no alterations in the envenomation process and mortality. The $ED_{50}$ was expressed in microliters. Prism (GraphPad Inc., San Diego, CA) was used to calculate the data by non-linear regression. Experiments were carried out following the guidelines published by WHO involved in the production, control, and regulation of venoms and antivenoms[54].

**Antiserum titration**. Each horse hyperimmune serum was tested by ELISA for the presence of specific antibodies against the ScNtx. Flat- bottom, 96-MicroWell™ polystyrene microtiter plates (Maxisorp Nunc) were coated with 100 μL/well of 5 μg/mL ScNtx in carbonate/bicarbonate stock solution at pH 9.5 and incubated overnight at 4 °C. After incubation, plates were rinsed three times with 200 μL/well of rinsing buffer (50 mM Tris-HCl, 150 mM NaCl, 0.05% Tween-20, and pH 8). After that, it was added 150 μL/well of blocking buffer (50 mM Tris-HCl, 5 mg/mL gelatin, 0.2% Tween-20, and pH 8) and incubated 2 h at 37 °C. After a second rinsing cycle, serum anti-ScNtx (1:30 dilution) was mixed with vehicle buffer (50 mM Tris-HCl, 0.5 M NaCl, 1 mg/mL gelatin, 0.05% Tween-20, and pH 8), and placed in the first well (200 μL) and further serially diluted 1:3 (with the same buffer of the ELISA plates) from wells 2 to 11, while well 12 contained only vehicle buffer. Plates were incubated for 1 h at 37 °C, and after rinsing twice; plates were incubated with 100 μL/well of peroxidase-conjugated rabbit anti-horse immunoglobulins ($5 \times 10^{-4}$ μg/mL, Rockland) for 1 h at 37 °C. Plates were then rinsed three times with rinsing buffer and, finally, 100 μL/well of peroxidase chromogenic substrate (soluble BM Blue POD substrate, Roche) were added, and incubated for 10 min at room temperature, in darkness. At the end of the incubation, the reaction was stopped with 100 μL/well sodium dodecyl sulfate 5%. The absorbance of the plates was read at 450 nm and the results were plotted using Prism 4.0 graphic package with non-linear analysis of regression[51].

**Reporting summary**. Further information on research design is available in the Nature Research Reporting Summary linked to this article.

## Data availability
The authors declare that the data that support the findings are included in the paper. Additional information is available from the corresponding author(s) upon request.

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

## Acknowledgements

We acknowledge Ricardo Mondragon, M. en C. Alejandro Olvera, M. en Biotec. Herlinda Clement, Dr. Bradley Yates, and personal from "Ojo de Agua" Ranch and Instituto Clodomiro Picado for providing technical support. This work received funding from the Direccion General de Asuntos del Personal Academico (DGAPA-UNAM) grant number IN203118 awarded to G.C. and IN207218 awarded to A.A. Partial financial support by Vicerrectoria de investigacion, Universidad de Costa Rica (VI-B7608), awarded to B.L. is gratefully acknowledged.

## Author contributions

G.d.l.R. performed Lot #1 and the other experiments and wrote the manuscript. I.G.A. performed the second Lot #2 of experiments. F.O. performed the purification of IgGs. A.A. and B.L. contributed to the study concept, design, and sample acquisition. G.d.l.R. and G.C. conceived and designed the project. G.C. served as principal investigator. All authors discussed the results, contributed to critical revisions, and approved the final manuscript.

## Additional information

**Competing interests:** The authors declare no competing interests.

