## [Peer Review File · Nature Communications]

Reviewers' comments:

Reviewer #1 (Remarks to the Author):

The manuscript entitled, *One toxin to rule them all: a consensus α -neurotoxin to improve antivenoms*, describes a highly exciting and promising study, which for the first time properly demonstrates the feasibility of using consensus toxins as immunogens for antivenom development. The findings are worthy of publication, however, the manuscript has a number of shortcomings that should be addressed.

1. Explanation of findings:

The neutralization results observed for certain snake venoms are not properly explained in the discussion. As an example, the paper Lauridsen, Line P., et al. "Toxicovenomics and antivenom profiling of the Eastern green mamba snake (*Dendroaspis angusticeps*)." *Journal of proteomics* 136 (2016): 248-261. clearly demonstrates that Eastern green mamba does NOT contain alpha-neurotoxins, yet in line 248 (and in the surrounding discussion) it is presented as if the antivenom neutralizes short neurotoxins of green mamba venom (which clearly cannot be the case). Similar incorrect/imprecise statements are made for several different snake venoms, where the authors neglect to relate their findings to what is known about the venom compositions from venomics studies.

It is recommended to carefully relate all findings more stringently to what has previously been described in the literature (particularly venomics studies).

An easy way of finding the relevant studies to refer to is via this database that links to all the original research articles: <http://tropicalpharmacology.com/tools/snake-venomics-display/relative-abundance-display/>

2. A number of relevant papers in the literature seems relevant to cite. Particularly in relation to discussing the importance of key toxins vs. other components, which is discussed in the following two papers:

Laustsen, Andreas H., et al. "Selecting key toxins for focused development of elapid snake antivenoms and inhibitors guided by a Toxicity Score." *Toxicon* 104 (2015): 43-45.

Laustsen, Andreas Hougaard. "Toxin synergism in snake venoms." *Toxin Reviews* 35.3-4 (2016): 165-170.

Particularly, the latter of these may provide hints to why green mamba venom is neutralized, if one speculates that the antivenom is cross-reactive towards either muscarinic 3FTx or synergistically-acting toxin.

3. In regards to the discussion on conserved epitopes in lines 147-148 it seems quite relevant to relate the findings to the following paper, which discusses this on a broad scale for mambas:

Engmark, Mikael, et al. "High-throughput immuno-profiling of mamba (*Dendroaspis*) venom toxin epitopes using high-density peptide microarrays." *Scientific reports* 6 (2016): 36629.

4. Cross-reactivity should be more elegantly discussed and elaborated (preferably in connection with reported venom compositions of the different snake venoms included in the study). What is the basis of the observed cross-reactivity? It seems that the antivenom not only recognizes short neurotoxins, but also long neurotoxins, as well as possibly 3FTxs in general.

5. It would be relevant to better relate the current findings to prior art. A comprehensive overview of all prior art in innovative immunization strategies can be found here: Bermúdez-Méndez, Erick, et al. "Innovative Immunization Strategies for Antivenom Development." *Toxins* 10.11 (2018): 452.

Additionally, the manuscript should be thoroughly proofread. Selected examples:

Line 49: designing => design

Line 72: THE market

Line 95: an => a "universal" (it is the pronunciation that counts, not that universal starts with a u - pronounced "ju")

Line 100: to induce => in inducing

Line 107: Effectiveness => efficacy

Line 107-108: Reads very poorly

Line 112: add "the" in front of Micrurus

The entire section entitled "ScNTx generates a powerful antivenom" is very poorly written. Both obvious grammatical errors should be corrected, as well as the language should be improved. There are many dozens of incorrect use of words and sentences that read poorly.

Other imprecisions

Line 100: "approx. 15,000" I cannot see how you came to this number from the figure?!?

Line 131: Please add reference for caprylic acid prec.

Line 132: It is stated that the antivenom contains 47 mg/mL protein, whereas throughout the rest of the manuscript the number 50 is used. Very confusing. Please correct

Line 139: Suddenly abbreviations for recombinant toxins are introduced. I have no idea what these are. Please specify better. I could not even find it in the methods.

Line 174: Add reference

Line 211: Remove 1 "naja"

Line 215: What is a protagonist agent? Find a better word (even if it is correct, it is highly confusing)

Line 224: a-cobratoxin is not a type I neurotoxin, but a type II. Please read the literature you cite properly

Line 247: "panorama"?!?

Line 278: "synthetic, recombinant". Here, the word "synthetic" is misleading, as the toxin is not chemically synthesised, but instead recombinant. If the word "artificial, recombinant" (or just unnatural or consensus) toxin was used, it would be better.

I wish you the best of luck and hope to see this paper published, as soon as the discussion has been improved according to the points raised.

Best regards

Andreas H. Laustsen

Reviewer #2 (Remarks to the Author):

General comments

Potentially an interesting article which is poorly written and marred by a lack of flow. These results is confusion to the reader. The English grammar is also in need of attention throughout the manuscript. The premise of the manuscript is that short chain alpha neurotoxins are primarily responsible for envenoming in humans. However, a recent paper (Silva et al. Cellular and Molecular Life Sciences; doi: 10.1007/s00018-018-2893-x) suggests that long chain alpha neurotoxins may be more important and that there are marked species differences in susceptibility which makes extrapolation from animal data difficult.

The Australian elapids (which are thought to be the most venomous snakes in the world) are not included in the study.

Specific comments

The statement that '...short-chain alpha-neurotoxins....play a major role in the envenomation process' (abstract) needs to be strengthened by further references. Also the statement that '...Post synaptic alpha-neurotoxins are one of the main toxic elements in elapid venoms and the most poorly recognized components by current antivenoms' (page 3, lines 53-56) (which is supported by a single 2011 paper on one snake venom).

I have similar concerns regarding the statement on page 4, lines 91-92.

Page 3, line 53. The difference between 'efficacy' and 'effectiveness' of antivenoms needs to be more clearly articulated. I think the authors have then incorrectly used 'effectiveness' throughout their manuscript when they are, in fact, examining efficacy.

Page 5, line 123. How were these time periods chosen?

Can the authors provide further insights into the wide variation seen in Figure 1?

May 1, 2019

Prof. Tanya Bondar, PhD
Senior Editor,
Nature Communications.

Dear Prof. Bondar

The authors of the submitted work **NCOMMS-18-35832-T** kindly thank the reviewers for their comments, which have allowed the improvement of the manuscript. Now, the manuscript has been enhanced and also carefully prepared for the resubmission process.

Here it is our response.

Reviewers' comments:

Reviewer #1 (Remarks to the Author):

The manuscript entitled, One toxin to rule them all: a consensus α -neurotoxin to improve antivenoms, describes a highly exciting and promising study, which for the first time properly demonstrates the feasibility of using consensus toxins as immunogens for antivenom development. The findings are worthy of publication; however, the manuscript has a number of shortcomings that should be addressed.

1. Explanation of findings:

The neutralization results observed for certain snake venoms are not properly explained in the discussion. As an example, the paper Lauridsen, Line P., et al. "Toxicovenomics and antivenom profiling of the Eastern green mamba snake (*Dendroaspis angusticeps*)." *Journal of proteomics* 136 (2016): 248-261. clearly demonstrates that Eastern green mamba does NOT contain alpha-neurotoxins, yet in line 248 (and in the surrounding discussion) it is presented as if the antivenom neutralizes short neurotoxins of green mamba venom (which clearly cannot be the case). Similar incorrect/imprecise statements are made for several different snake venoms, where the authors neglect to relate their findings to what is known about the venom compositions from venomics studies.

Answer: The article of Lauridsen et al. (2016) shows in table No. 1 a protein fraction (8a) with a peptide ion of 1,685.8 having a predicted MS/MS sequence of **AILTNCGENSCYRK**, and it belongs to a 3FTx (fasciculin/Ta2) found and described by Viljoen and Botes (1974). Furthermore, peak 8 was the only toxic fraction reported by Lauridsen et al. (2016) with an LD₅₀ of 0.58 mg/kg (CI, 0.17–1.23). So, fasciculins display lethal activity in mice and they share 35% amino acid sequence identity to ScNTx, this identity perhaps is enough to be recognized by anti-ScNtx antibodies. Also, anti-ScNtx antibodies may also recognize 3D structures, both ScNtx and Ta2

share similar cysteine scaffold, and it will be not surprise that anti-ScNtx antibodies recognize also other 3FTxs, toxic or not.

```
ScNtx  MICYNQQSSQPPTTKTCSETSCYKKTWRDHRGTIIEGCGCPKVKPGIKLHCERT-DKCNN
Ta2    TMCYSHTTTSRAILTNCGENSCYRKSRRHPPKMVLGRGCGCPPGDDNLEVKCCTSPDKCNY
      :*: : :  ..*.***:*:* : : *****  .:***:****
```

In addition to the possible cross-reactivity of the experimental antivenom to ScNtx with 3FTxs in green mamba venom that are not defined as "neurotoxins" (such as the fasciculin example presented above), it is also reasonable that proteomic analysis of this venom might have missed the detection of one or more classical type I neurotoxins if the case was that these have never been sequenced and deposited in databases (protein identification in proteomics is largely data-dependent). In other words, venom proteomics is not the absolute final method to define the total protein composition of a biological sample, and indeed the paper by Lauridsen, Line P. et al. reports an "Unknown" fraction of the venom which represents 4.9%. In our view, the result obtained in the neutralization assay of green mamba venom has to be objectively reported "as is", regardless of its possible explanations or rationalizations (which we agree will be interesting to be investigated in studies to follow). But it seems difficult for us to argue that the result "clearly cannot be the case", as indicated by the reviewer, considering that several possible explanations could be involved in this finding.

It is recommended to carefully relate all findings more stringently to what has previously been described in the literature (particularly venomics studies).

Answer: The reviewer is correct. Venomics and antivenomic studies are becoming an important technology to study the composition of animal venoms, but also it has some weaknesses if they are not supported by in vivo studies. Unfortunately, few venomics studies are correlated to in vivo experimental analysis, so it is difficult to find literature with both type of studies.

An easy way of finding the relevant studies to refer to is via this database that links to all the original research articles: <http://tropicalpharmacology.com/tools/snake-venomics-display/relative-abundance-display/>

Answer: The database was consulted, thanks; so, most of the venoms used in this work such *Dendroaspis polylepis*, *Micrurus nigrocinctus*, *Micrurus fulvius*, *Naja atra* (China/Taiwan), *Naja kaouthia* (Malaysia/Thailand/Vietnam), *Naja melanoleuca*, *Naja mossambica*, *Naja naja* (East India/Northwest India/Sri Lanka) *Naja nigricollis*, *Naja nubiae*, *Naja pallida*; and *Hydrophis platura* contain short-chain neurotoxins. It seems that *Dendroaspis angusticeps* does not contain short-chain neurotoxins; but it has other toxic 3FTxs (fasciculins Ta1/Ta2, Viljoen and Botes, 1974; please see answer to point 1 above).

2. A number of relevant papers in the literature seems relevant to cite. Particularly in relation to discussing the importance of key toxins vs. other components, which is discussed in the following

two papers:

Laustsen, Andreas H., et al. "Selecting key toxins for focused development of elapid snake antivenoms and inhibitors guided by a Toxicity Score." *Toxicon* 104 (2015): 43-45.

Laustsen, Andreas Hougaard. "Toxin synergism in snake venoms." *Toxin Reviews* 35.3-4 (2016): 165-170.

Answer: We thank the reviewer for this observation, and accordingly we have included a reference that we consider relevant: *Toxicon* 104 (2015): 43-45. For the revised version see reference 11.

Particularly, the latter of these may provide hints to why green mamba venom is neutralized, if one speculates that the antivenom is cross-reactive towards either muscarinic 3FTx or synergistically-acting toxin.

Answer: Concerning synergistically-acting toxins, it is clear that most animal venoms (vg. arachnids, elapids, viperids, cone snails) contain venom components that potentiate each other; however, they are few critical toxic peptide molecules that affect humans such Nav modifiers in scorpion and some spider venoms; sphingomyelinases and latrotoxins in venoms from *Loxocles* and *Lactrodectus* spiders, respectively; ACh blockers or AChase inhibitors from elapids, all of them are the leading toxic compounds that if they are neutralize such animal venoms become less toxic, and also the synergistically-acting effect decreases. Respect to cross-reactive, in the case of this work, it is possible that antibodies against ScNtx may recognize also either muscarinic 3FTx, or other synergistically-acting toxins, but it has to be proven. Our guest, according to the primary structures of short-chain neurotoxins, is that their core is highly conserved, and the antibodies against ScNtx recognize the short-chain neurotoxins from elapid venoms (and it may recognize fasciculins from *Dendroaspis angusticeps*). We mentioned this throughout the text.

3. In regards to the discussion on conserved epitopes in lines 147-148 it seems quite relevant to relate the findings to the following paper, which discusses this on a broad scale for mambas: Engmark, Mikael, et al. "High-throughput immuno-profiling of mamba (*Dendroaspis*) venom toxin epitopes using high-density peptide microarrays." *Scientific reports* 6 (2016): 36629.

Answer: *Dendroaspis* species and selected African *Naja* species was performed based on custom-made high-density peptide microarrays displaying linear toxin fragments. By detection of binding for three different antivenoms and performing an alanine scan, linear elements of epitopes and the positions important for binding were identified. Nonetheless, authors agree to relate our findings to Engmark et al; therefore, the reference was included, see lines 145 and *Reference No 14*

4. Cross-reactivity should be more elegantly discussed and elaborated (preferably in connection with reported venom compositions of the different snake venoms included in the study). What is the basis of the observed cross-reactivity? It seems that the antivenom not only recognizes short neurotoxins, but also long neurotoxins, as well as possibly 3FTxs in general.

Answer: As mentioned cross-reactivity exists among venom species of the same family from different regions such as scorpions, spiders (*Loxocles/Latrodectus*), elapids and viperids. Concerning short- and long- chain neurotoxins, although they are quite different in their primary structures, perhaps only the large amounts of cationic charged peptides (they are immunogenic residues) in them may be responsible for cross reactivity, but it has to be proven. In fact, short-chain neurotoxins are slightly more toxic than long-chain neurotoxins, and also according to some reports the concentration of short-chain neurotoxins are higher than that of long-chain neurotoxins for many elapid venoms (please see Weinstein et al. 1991, Lethal toxins and cross-neutralization of venoms from the African water cobras, *Boulengerina annulata annulata* and *Boulengerina christyi*. *Toxicon* 11:1315-27, and Tan et al. 2017, Venomics of *Naja sputatrix*, the Javan spitting cobra: A short neurotoxin-driven venom needing improved antivenom neutralization. *J Proteomics* 157:18-32). If so, the antibodies that recognize *ScNTx* would neutralize the large concentration of short-chain neurotoxins, and in some extent recognize also other 3TXs including fasciculins (35% identity), long- chain neurotoxins (43% identity) and Cytotoxin/cardiotoxins (>40%). We considered important the reviewer's comment. In the revised manuscript, authors have incorporated cross-reactivity as an important factor for discussion.

```
Bgtx-A31 IVCHT-TATSPISAVTCPPGENLCYRKMWCDAFCSSSRGKVVELGCAATCPSKPPYEEVTCST-DKCNPHPKQRPG
ScNtx MICYNQQSSQPPTTKTC--SETSCYKKTWRD----HRGTIIERGCG--CPKVKPGIKLHCCRT-DKCNN-----
:::.. ::: : ** . **:* ** **::* ** **:* ** *****
```

5. It would be relevant to better relate the current findings to prior art. A comprehensive overview of all prior art in innovative immunization strategies can be found here: Bermúdez-Méndez, Erick, et al. "Innovative Immunization Strategies for Antivenom Development." *Toxins* 10.11 (2018): 452.

Answer: Innovative immunization strategies of course are interesting, the use of synthetic peptide epitopes, or DNA strings as immunogens have been demonstrated for generating antibodies. However, it is important to scale up such results to large animals (the ones that are used to develop current antivenoms). The focus of our work was the proof of concept of a recombinant toxin with consensus sequence that could become cost-effective for traditional antivenom producers; now, it is an important issue that antivenom producers have to consider. As far as we know, the only innovative immunization strategy that has been used commercially nowadays is the recombinant sphingomyelinase D for anti-brown spider venom.

Additionally, the manuscript should be thoroughly proofread. Selected examples:

Line 49: designing => design. Answer: it was amended.

Line 72: THE market .Answer: it was amended.

Line 95: an => a "universal" (it is the pronunciation that counts, not that universal starts with a u - pronounced "ju"). Answer: it was amended.

Line 100: to induce => in inducing. Answer: it was amended.

Line 107: Effectiveness => efficacy. Answer: it was amended.

Line 107-108: Reads very poorly. Answer: it was amended.

Line 112: add "the" in front of *Micrurus*. Answer: it was amended.

The entire section entitled "*ScNTx* generates a powerful antivenom" is very poorly written. Both

obvious grammatical errors should be corrected, as well as the language should be improved. There are many dozens of incorrect use of words and sentences that read poorly.

Answer: Thank you, the manuscript was read by a native speaker, and grammatical improved.

Other imprecisions

Line 100: "approx. 15,000" I cannot see how you came to this number from the figure?!?

Line 131: Please add reference for caprylic acid prec. **Answer: thank you, the corresponding reference was added. See references # 13, 57 and 58.**

Line 132: It is stated that the antivenom contains 47 mg/mL protein, whereas throughout the rest of the manuscript the number 50 is used. Very confusing. Please correct **Answer: thank you, it was corrected.**

Line 139: Suddenly abbreviations for recombinant toxins are introduced. I have no idea what these are. Please specify better. I could not even find it in the methods.

Line 174: Add reference. **Answer: thank you, it was added.**

Line 211: Remove 1 "naja" **Answer: thank you, it was removed**

Line 215: What is a protagonist agent? Find a better word (even if it is correct, it is highly confusing) **Answer: thank you, it was amended.**

Line 224: a-cobratoxin is not a type I neurotoxin, but a type II. Please read the literature you cite properly. **Answer: thank you, it was corrected. It is true that a-cobratoxin is a type II.**

Line 247: "panorama"?!? **Answer: it was changed by "view"**

Line 278: "synthetic, recombinant". Here, the word "synthetic" is misleading, as the toxin is not chemically synthesised, but instead recombinant. If the word "artificial, recombinant" (or just unnatural or consensus) toxin was used, it would be better. **Answer: Thank you, now we used the term "recombinant"**

I wish you the best of luck and hope to see this paper published, as soon as the discussion has been improved according to the points raised.

Best regards

Andreas H. Laustsen

Reviewer #2 (Remarks to the Author):

General comments:

Potentially an interesting article which is poorly written and marred by a lack of flow. These results is confusion to the reader. The English grammar is also in need of attention throughout the manuscript.

Answer: Thank you for your observations. The English grammar was revised by a native English speaker, we hope you will now find it suitable.

The premise of the manuscript is that short chain alpha neurotoxins are primarily responsible for

envenoming in humans. However, a recent paper (Silva et al. Cellular and Molecular Life Sciences; doi: 10.1007/s00018-018-2893-x) suggests that long chain alpha neurotoxins may be more important and that there are marked species differences in susceptibility which makes extrapolation from animal data difficult.

Answer: As the reviewer mentioned, Silva *et al.* suggest that long chain alpha neurotoxins are more important than short chain alpha neurotoxins, but the work was based on in vitro experiments (electrophysiology) where other biological variables are dismissed such, and most importantly there is no *in vivo* proof supporting that long chain alpha neurotoxins are more toxic. In literature, the LD₅₀ determined (by I.V.) for short-chain neurotoxins are on average three-fold more toxic than long-chain neurotoxins. Therefore, we considered that is difficult to extrapolate the behavior in-vitro of long-chained neurotoxins to in vivo.

The Australian elapids (which are thought to be the most venomous snakes in the world) are not included in the study.

Answer: The reviewer is correct, our stock of venoms did not have Australian venoms; However, in the revised version and in a new set of experiments we incorporated three Australian venoms.

Specific comments

The statement that '...short-chain alpha-neurotoxins....play a major role in the envenomation process' (abstract) needs to be strengthened by further references. Also the statement that '...Post synaptic alpha-neurotoxins are one of the main toxic elements in elapid venoms and the most poorly recognized components by current antivenoms' (page 3, lines 53-56) (which is supported by a single 2011 paper on one snake venom).

Answer: We appreciate this comment. We support this statement by using different references throughout the text. To mention some: *Laustsen et al. (2015) J Proteomics 119, 126-42, determining the "toxicity scores" for black mamba toxins, demonstrates that short-chained toxins are the most toxic.* So was the case for references # 5, 4, 14, 18, 24, 27, 31, 32.

I have similar concerns regarding the statement on page 4, lines 91-92.

Page 3, line 53. The difference between 'efficacy' and 'effectiveness' of antivenoms needs to be more clearly articulated. I think the authors have then incorrectly used 'effectiveness' throughout their manuscript when they are, in fact, examining efficacy.

Answer: We addressed such comment, and in agreement with the reviewer, we amended and clarify such concept. Thank you.

Page 5, line 123. How were these time periods chosen?

Answer: In literature is recommended monitoring the rise in neutralizing potency as a result of specific antibodies against the immunogen (see below). The periods were chosen according to the antivenom manufacturer company (Inosan biopharma, <http://inosanbiopharma.com>), which protocols are used for several commercially available antivenom products.

Can the authors provide further insights into the wide variation seen in Figure 1?

Answer: The immune system of complex organism like mammals has a high degree of variation between individuals. There are drivers of immune variation such as age, stress, gender, or any particular stimulus at any given moment. This variation is also present in the antigen receptor of adaptive immune system (T cell receptor and B cell receptor) that ultimately impacts the humoral response (antibodies). This variation has also been reported for rabbits (de la Rosa et al., *Toxicon* **155**, 32-37 (2018)) and mention for horses (León, G. et al., *Toxicon* **151**, 63–73 (2018)) immunized with toxins or whole venoms.

REVIEWERS' COMMENTS:

Reviewer #1 (Remarks to the Author):

The manuscript has significantly improved and is now almost ready for publication/acceptance.

The main finding/conclusion that recombinant consensus toxins can be used as immunogens for producing broadly neutralizing snakebite antivenoms is novel and highly encouraging. This paper is expected to be of wide interest to the field of toxinology and possibly also the fields of vaccinology and applied immunology.

Methods, analyses, and results are relevant and clearly described. Claims are now better presented and supported with reference to appropriate prior art.

Minor comments:

Commas missing. I also suggest using Oxford comma consistently

Spaces either missing (such as after LD50 or ED50 several places and after toxin family abbreviations) or are too many throughout the manuscript (sometimes before references)

Use subscript for 2 for PLA2 everywhere

Line 102: Should titer have a unit?

Line 118: Outstanding is not a very descriptive word. Please change to a more descriptive word, such as broadly specific or cross-reactive

Line 151: pathophysiological is misspelled

Some places uniprot protein IDs are italicized, other places not. Be consistent

Andreas H. Laustsen

Reviewer #2 (Remarks to the Author):

The authors have amended their manuscript in line with the majority of my comments. I am pleased to see the the venoms of some Australasian elapids have been added.

The grammar is still problematic in parts but I'm hoping that this may be detected (and corrected) in the next phase.

The authors have missed (and largely dismissed) my point about the importance of short- and long-chain post synaptic neurotoxins in HUMAN envenoming. The study by Silva et al used HUMAN receptors. This is not a matter of in vitro versus in vivo (as hypothesised by the authors of the current manuscript). The LD50 studies in the current study were undertaken in MICE (rodents). The N receptors in mice are clearly different from the human receptors.

The final paragraph of the Silva paper states "The differential susceptibility of human and rat nAChR to snake SaNTx may also be PROBLEMATIC FOR LETHALITY ASSAYS IN RODENTS TO ASSESS ANTIVENOM EFFICACY. Rodent lethality and ED50 studies of cobra venoms that only contain SaNTx, WILL NOT MODEL HUMAN ENVENOMING.....THEREFORE, RODENT MODELS MAY BE IRRELEVANT FOR TESTING ANTIVENOM'S CLINICAL EFFECTIVENESS'. The authors need to address these concerns in the manuscript rather than dismissing the concerns.

REVIEWERS' COMMENTS:

Reviewer #1 (Remarks to the Author):

The manuscript has significantly improved and is now almost ready for publication/acceptance.

Answer: Thank you for your frank opinion.

The main finding/conclusion that recombinant consensus toxins can be used as immunogens for producing broadly neutralizing snakebite antivenoms is novel and highly encouraging. This paper is expected to be of wide interest to the field of toxinology and possibly also the fields of vaccinology and applied immunology.

Answer: Thank you, we think that the use of consensus recombinant neurotoxins/enzymes, from animal venoms, as immunogens will bring new concepts for antivenom production.

Methods, analyses, and results are relevant and clearly described. Claims are now better presented and supported with reference to appropriate prior art.

Answer: Thank you very much.

Minor comments:

Commas missing. I also suggest using Oxford comma consistently.

Answer: Commas are now used consistently.

Spaces either missing (such as after LD50 or ED50 several places and after toxin family abbreviations) or are too many throughout the manuscript (sometimes before references)

Answer: Spaces between were introduced where they were missing

Use subscript for 2 for PLA₂ everywhere

Answer: Now all PLA abbreviations have the subscript 2

Line 102: Should titer have a unit?

Answer: An antibody titer is dimensionless, in this case IgG/mL:IgG/mL expressed as the inverse of the greatest dilution (in a serial dilution), so in figure 1, the units are from 0 to 1:21,000 (IgG/mL:IgG/mL).

Line 118: Outstanding is not a very descriptive word. Please change to a more descriptive word, such as broadly specific or cross-reactive

Answer: The word outstanding was replaced

Line 151: pathophysiological is misspelled

Answer: The word physiopathological was replaced for pathophysiological

Some places uniprot protein IDs are italicized, other places not. Be consistent

Answer: Now all uniprot protein IDs are not italicized

Andreas H. Laustsen

Reviewer #2 (Remarks to the Author):

The authors have amended their manuscript in line with the majority of my comments. I am pleased to see the the venoms of some Australasian elapids have been added.

Answer: Thank you very much.

The grammar is still problematic in parts but I'm hoping that this may be detected (and corrected) in the next phase.

Answer: The grammar was once more revised for a Canadian English speaker, hopefully is according to the standard commonwealth English.

The authors have missed (and largely dismissed) my point about the importance of short- and long-chain post synaptic neurotoxins in HUMAN envenoming. The study by Silva et al used HUMAN receptors. This is not a matter of in vitro versus in vivo (as hypothesised by the authors of the current manuscript). The LD50 studies in the current study were undertaken in MICE (rodents). The N receptors in mice are clearly different from the human receptors.

The final paragraph of the Silva paper states "The differential susceptibility of human and rat nAChR to snake S α NTx may also be PROBLEMATIC FOR LETHALITY ASSAYS IN RODENTS TO ASSESS ANTIVENOM EFFICACY. Rodent lethality and ED50 studies of cobra venoms that only contain S α NTx, WILL NOT MODEL HUMAN ENVENOMING.....THEREFORE, RODENT MODELS MAY BE IRRELEVANT FOR TESTING ANTIVENOM'S CLINICAL EFFECTIVENESS'. The authors need to address these concerns in the manuscript rather than dismissing the concerns.

Answer: We apologize for missing such point concerning the importance of short- and long-chain post synaptic neurotoxins in human envenoming. We read the interesting article from Silva et al. article, and we discuss the following:

As pointed out by Silva et al. (2018), there is no doubt that long-chain α -neurotoxins plays and important role in elapid envenomation in Africa/Asia/Australia. Furthermore, our work de la Rosa et al. (2016) communicates that synthetic peptide antigens derived from long-chain alpha-neurotoxins generates partially-neutralizing antibodies, which delayed mice mortality in neutralization assays against *Naja haje*, *Dendrospis polylepis* and *Ophiophagus hannah* venoms. However, the fact that the experimental anti-ScNTx could neutralize some *Micrurus sp* and *Naja sp* venoms means that there is some cross-reactivity neutralization among elapid venoms. An example is the work of Wisniewski et al. (2003), who reports that the tiger snake (*Notechis scutatus*) and Mexican coral snake (*Micrurus* species) antivenoms prevent death from United States coral snake (*Micrurus fulvius fulvius*), all these neutralization assays were performed in mice as a model. Furthermore, Paniagua et al. (2012 and 2019) have studied the pharmacokinetics of *Micrurus sp.* venoms in medium size animals (vg sheep) rather than only in rodents, avian and amphibians (as mentioned in Silva et al. 2018). *Micrurus sp.* (coral snakes) are American elapids from North (USA) to South (Argentina) America that contains only short α -neurotoxins in their venoms. They found that anti-coral snake venoms currently used to save humans envenomed by *Micrurus* coral snakes abolish sheep envenomation, and also they use mice follow in vivo neutralization, and the quality of such antivenoms. Actually, in America four countries produce commercial anti-coral snake venoms, that contains short α -neurotoxins, for protection of their population (Instituto Bioclon, Mexico; Instituto Clodomiro Picado, Costa Rica; Fundação Ezequiel Dias (FUNED), Brazil; Instituto Butantan, Brazil; Instituto Nacional de Producción de Biológicos (ANLIS) "Dr Carlos Malbrán", Argentina), please see Gutierrez (2018) Preclinical assessment of the neutralizing efficacy of snake antivenoms in Latin America and the Caribbean: A review. Toxicon 146:138-150. All those research centers that produce coral snake antivenoms use mice as a model for keeping antivenom quality, as following WHO recommendations. Perhaps pharmacokinetics using monoclonal antibodies against either short- or long-chain alpha-neurotoxins during envenomation of animals could shed light to this issue.

A paragraph indicating the importance of long- and short-chain alpha-neurotoxins is included in the main text:

“Here, it is worth mentioning that long- and short-chain α -neurotoxins are equally important during elapid envenomation, and more studies are needed to decipher the pharmacokinetics and biochemistry of each one of these neurotoxins. Also, the use of mice as an animal model for interpreting neutralization of rich long- and rich short-chain α -neurotoxin venoms has to be revised. Silva et al. (2018) found differential susceptibility of human and rat nAChR to short-chain α -neurotoxins, but not to long-chain α -neurotoxins. So, this fact may be challenging for assessing clinical antivenom efficacy in mice.”